# Effect of Abutment Screw Design on Torque Loss Under Cyclic Fatigue Loading: A Comparison of TSIII and KSIII Implant Systems

**DOI:** 10.3390/bioengineering12101131

**Published:** 2025-10-21

**Authors:** Jung-Tae Lee, Jae-Chang Lee, Dong-Wook Han, Bongju Kim

**Affiliations:** 1Department of Periodontics, One-Stop Specialty Center, Seoul National University Dental Hospital, Seoul 03080, Republic of Korea; jungtae1308@hanmail.net; 2Research Center for Bio-Based Chemistry, Korea Research Institute of Chemical Technology (KRICT), Ulsan 44429, Republic of Korea; jclee@krict.re.kr; 3Department of Cogno-Mechatronics Engineering, Pusan National University, Busan 46241, Republic of Korea; nanohan@pusan.ac.kr; 4Dental Life Science Research Institute, Seoul National University Dental Hospital, Seoul 03080, Republic of Korea

**Keywords:** dental implants, abutment screw loosening, preload, torque loss, fatigue loading, implant–abutment connection, biomechanics

## Abstract

Background: Abutment screw loosening (ASL) is the most frequent mechanical complication in dentistry, leading to prosthetic instability and biological risks. Preload, generated during screw tightening, is critical for maintaining stability but is influenced by torque application, screw geometry, and cyclic loading. Methods: This in vitro study compared torque loss between two implant systems (Osstem TSIII and KSIII) with different abutment screw designs. Fifty implant–abutment assemblies (*n* = 5 per torque group) were tested under tightening torques of 20, 25, 30, 35, and 40 Ncm. Initial removal torque (T1) was measured 5 min after tightening, followed by cyclic loading (150 N, 14 Hz, 100,000 cycles). Post-fatigue removal torque (T2) was then recorded, and torque loss rate (%) was calculated. Independent *t*-tests and a one-way ANOVA were used for statistical analysis. Results: KSIII consistently exhibited higher T1 and T2 values than TSIII across all torque levels (*p* < 0.05). The torque loss rate for TSIII ranged from 36.5% (35 Ncm) to 51.8% (20 Ncm), showing a torque-dependent trend (*p* < 0.05). In contrast, KSIII maintained torque loss rates below 25% at all levels, with no significant differences between torque groups (*p* > 0.05). On average, torque loss in TSIII was approximately 2.5–3.0 times higher than in KSIII. Conclusions: The KSIII system demonstrated superior biomechanical stability, with significantly lower torque loss compared with TSIII, independent of torque level. Clinically, these findings suggest that the KSIII system may reduce the incidence of screw loosening and associated complications. A tightening torque of approximately 35 Ncm appeared to provide the most stable preload. Long-term in vivo studies are warranted to confirm these results under clinical conditions.

## 1. Introduction

Dental implants are widely recognized as a predictable and effective treatment modality for the replacement of missing teeth, with cumulative survival rates reported above 90% over 10 years of function [1,2]. Despite their high success, both biological and mechanical complications still occur and can undermine treatment outcomes [3]. Among these, abutment screw loosening (ASL) is consistently identified as the most frequent mechanical complication, with incidence rates ranging from 5% to 15% depending on implant system, loading conditions, and follow-up duration [4,5]. Clinically, screw loosening compromises prosthesis stability, creates microgaps at the implant–abutment interface, and facilitates bacterial leakage, which may lead to peri-implant mucositis or peri-implantitis [3]. Recurrent loosening episodes further risk component fracture, difficulty in retrieval, and ultimately implant failure [1].

The stability of an abutment screw depends largely on the preload, defined as the tensile force generated when torque is applied during screw tightening [6]. Preload provides the clamping force that maintains intimate contact between the abutment and implant body, resisting functional occlusal and lateral forces [7]. However, only about 8–10% of applied torque is converted into preload, while the remaining energy is dissipated as friction at the screw head–abutment interface and along the screw threads [8]. This explains why minor differences in thread design, surface treatment, or lubrication can lead to significant differences in preload. Furthermore, the settling effect, also called embedment relaxation, occurs when microscopic surface asperities deform and flatten after initial tightening, leading to an immediate reduction in preload [9]. Haack et al. [8] reported that reverse torque values could decrease by 20–30% within minutes of tightening due to settling, highlighting the importance of preload management in clinical practice.

Previous research has explored various strategies to counteract preload loss. One commonly recommended protocol is retightening, in which the screw is tightened again after a short waiting period to compensate for the settling effect [10]. In vitro studies have shown that retightening within minutes after initial torque application significantly increases removal torque and reduces subsequent loosening [8,10,11]. Moreover, accurate implant component fit is also critical for long-term screw stability, as misfit can exacerbate preload loss. Studies on impression accuracy highlight that even small discrepancies at the implant–abutment level may affect preload transfer and stability [12]. In addition, screw material and surface modifications, such as carbon or gold plating, have been tested to improve frictional properties and enhance preload [13]. However, the effectiveness of these interventions is often implant system–dependent, and a universally accepted clinical protocol is still lacking.

Another critical factor is implant–abutment connection design. Early external hex connections, while widely used, have been reported to be more prone to rotational micromotion and screw loosening due to their shallow anti-rotational features [14]. In contrast, internal connections, particularly conical or Morse taper designs, exhibit superior resistance to torque loss and micromotion [15,16]. Synchrotron-based imaging has confirmed that while microgaps are present in all connection types, conical interfaces show more favorable deformation patterns that enhance stability under loading [17]. Moreover, angulated screw channel (ASC) abutments have been introduced for improved prosthetic flexibility and esthetics, but their altered screw access path may reduce effective preload transfer [18]. Clinical studies suggest that connection geometry significantly influences complication rates, underscoring its biomechanical importance [3].

Despite these advances, clinical reports continue to describe screw loosening. Retrieval analyses of failed screws revealed wear, deformation, and even corrosion, indicating that loosening is not solely mechanical but also involves tribological and material degradation processes [19,20]. Furthermore, environmental factors such as saliva, blood, and antimicrobial gels can alter friction coefficients at the interface, further influencing torque stability [21]. Taken together, these findings highlight the multifactorial nature of screw loosening and the need for implant systems designed to maintain preload more effectively.

Therefore, the aim of the present study was to compare the torque loss rates of abutment screws in two implant systems, Osstem TSIII and KSIII, across different tightening torques under cyclic fatigue loading. The null hypothesis was that no difference would be observed between the two systems, and it was tested against the alternative hypothesis that differences might exist.

## 2. Materials and Methods

### 2.1. Implant Systems and Abutments

Two commercially available dental implant systems (Osstem Implant Co., Ltd., Seoul, Republic of Korea) were selected for this study: TSIII and KSIII. Both systems were chosen due to their widespread clinical use and distinct abutment screw designs. The TSIII system employs a conventional abutment screw with a standard thread and head geometry, whereas the KSIII system utilizes a modified screw with a reduced head diameter, narrower thread profile, and increased nominal length, which are intended to improve preload stability. Cylindrical implants (Ø4.0 mm × 10 mm length) with corresponding straight abutments (Ø5.0 mm platform, 2.0 mm gingival height) were used (Figure 1). Both implant types share the same macro- and micro-design characteristics; the only difference between the two systems lies in the implant–abutment connection geometry. All fixtures were made of commercially pure titanium (Grade 4), and the abutments and abutment screws were composed of titanium alloy (Ti-6Al-4V).

### 2.2. Sample Size and Grouping

A total of 50 implant–abutment assemblies were prepared (*n* = 5 per group). The sample size was determined based on previous in vitro implant biomechanics studies, which commonly employed five specimens per group to balance experimental feasibility with statistical reliability [10,22]. Each system (TSIII and KSIII) was tested under five tightening torque conditions: 20, 25, 30, 35, and 40 Ncm. Thus, 10 groups were established (2 systems × 5 torque levels). All abutments and screws were supplied by the manufacturer and used without modification.

### 2.3. Torque Application and Measurement of Initial Removal Torque (T1)

Each implant–abutment assembly was embedded in a custom stainless-steel jig to ensure stable positioning during torque application. Each abutment screw was tightened to the assigned torque value using a calibrated digital torque gauge (MTT03-100, MARK-10, Long Island City, NY, USA). Torque was applied with a constant speed (approximately 1 rpm) to reduce variability, following previously established protocols for torque testing in implant biomechanics [8,23]. Five minutes after tightening, the initial removal torque (T1) was measured to evaluate preload loss due to the settling effect. All torque applications and removal torque measurements were performed by the same trained operator to ensure consistency and minimize inter-operator variability. All measurements were performed under controlled laboratory conditions (22 ± 1 °C, 50 ± 5% relative humidity).

### 2.4. Fatigue Loading Protocol

After the T1 measurement, each specimen was retightened to the assigned torque value to restore the initial preload. The implant–abutment assemblies were then embedded in a custom stainless-steel jig and mounted on a servo-hydraulic universal testing machine (ElectroPuls E3000, Instron, High Wycombe, UK) at the Dental Life Science Research Institute, Seoul National University Dental Hospital (Seoul, Republic of Korea). Dynamic cyclic loading was applied using a cantilever fatigue test protocol (Load: 150 N, Frequency: 14 Hz, Number of cycles: 100,000), in accordance with ISO 14801 guidelines [24] and previous studies on abutment screw loosening under cyclic loading [21,25].

This loading condition was selected to simulate one to two months of masticatory function in vivo. During testing, loading was applied at a 30° angle relative to the long axis of the implant to reproduce off-axis occlusal forces encountered in clinical function (Figure 2).

### 2.5. Measurement of Post-Fatigue Removal Torque (T2) and Torque Loss Rate

Immediately after completion of fatigue loading, the post-fatigue removal torque (T2) was recorded using the same digital torque gauge. While ISO 14801 provides standardized guidance for implant fatigue testing, it does not specifically address abutment screw torque measurement; therefore, our procedure was designed in accordance with previously published methods [8,23]. The torque loss rate (%) was calculated according to the following formula:Torque Loss Rate (%) = 100 × (*T*1 − *T*2)/*T*1

This parameter was used to compare preload maintenance capacity between implant systems at each torque level.

### 2.6. Statistical Analysis

All data were analyzed using statistical software (SPSS v26.0, IBM, Armonk, NY, USA). Normality of data distribution was assessed with the Shapiro–Wilk test. Independent *t*-tests were used to compare torque loss rates between the two systems (TSIII vs. KSIII) at each tightening torque. One-way ANOVA with post hoc Tukey’s test was applied to evaluate differences among torque levels within each system [10]. The level of statistical significance was set at *p* < 0.05.

## 3. Results

### 3.1. Initial Removal Torque (T1)

The mean values of initial removal torque (T1), measured 5 min after initial tightening, are presented in Table 1. Both TSIII and KSIII implant systems demonstrated a gradual increase in T1 with increasing tightening torque. At 20 Ncm, the TSIII group recorded a mean T1 of 16.4 ± 0.3 Ncm, whereas the KSIII group recorded 18.2 ± 0.3 Ncm. At 25 Ncm tightening, TSIII showed 19.3 ± 0.3 Ncm, while KSIII achieved 20.8 ± 0.3 Ncm. With further increments in torque, the differences remained consistent: at 30 Ncm, TSIII was 23.1 ± 0.7 Ncm compared to 27.1 ± 0.2 Ncm for KSIII; at 35 Ncm, TSIII was 30.3 ± 0.3 Ncm compared to 33.1 ± 0.3 Ncm for KSIII; and at 40 Ncm, TSIII showed 29.3 ± 1.1 Ncm while KSIII was 32.2 ± 0.6 Ncm.

Statistical analysis confirmed that the KSIII system consistently exhibited significantly higher T1 values than the TSIII system across all torque levels (independent *t*-test, *p* < 0.05 for all comparisons).

### 3.2. Post-Fatigue Removal Torque (T2)

After cyclic loading (150 N, 14 Hz, 100,000 cycles), removal torque values (T2) decreased in both implant systems, indicating preload loss under simulated functional conditions. The corrected results are summarized in Table 2.

At a tightening torque of 20 Ncm, the TSIII system decreased to 7.9 ± 0.2 Ncm, whereas the KSIII system maintained a significantly higher value of 14.0 ± 0.4 Ncm. At 25 Ncm, TSIII recorded 10.1 ± 0.5 Ncm compared with 17.4 ± 0.2 Ncm for KSIII. At 30 Ncm, TSIII retained 14.0 ± 0.6 Ncm, while KSIII reached 21.5 ± 0.4 Ncm. At 35 Ncm tightening, TSIII showed 19.2 ± 0.6 Ncm, and KSIII achieved 26.9 ± 0.5 Ncm. Finally, at 40 Ncm, TSIII exhibited 18.1 ± 0.7 Ncm, while KSIII maintained 24.7 ± 0.8 Ncm.

Statistical analysis (independent *t*-test) demonstrated that KSIII consistently preserved significantly higher post-fatigue torque values compared with TSIII across all torque levels. Intra-group analysis further revealed that in the TSIII system, T2 values showed a significant decreasing trend at lower torque levels, particularly between 20 and 35 Ncm (one-way ANOVA, *p* < 0.05). In contrast, the KSIII system maintained relatively stable T2 values across torque levels, and intra-group differences were not statistically significant (ANOVA, *p* > 0.05).

### 3.3. Torque Loss Rate (%)

The torque loss rate, calculated as the percentage reduction from the initial removal torque (T1) to the post-fatigue removal torque (T2), is summarized in Figure 3.

For the TSIII system, torque loss rates were markedly higher and showed a clear dependence on tightening torque. At 20 Ncm, the torque loss rate reached 51.8%, the highest among all groups. As tightening torque increased, the loss rate decreased: 47.7% at 25 Ncm, 39.4% at 30 Ncm, and 36.5% at 35 Ncm, representing the lowest value for TSIII. However, at 40 Ncm, torque loss slightly increased again to 38.2%, suggesting that excessive torque did not necessarily enhance preload maintenance. Statistical analysis confirmed significant differences among torque levels within the TSIII system (one-way ANOVA, *p* < 0.05), with particularly higher loss at 20 Ncm compared to 35 Ncm (Tukey post hoc, *p* = 0.008).

In contrast, the KSIII system consistently demonstrated substantially lower torque loss rates across all torque conditions. The lowest value was observed at 25 Ncm (16.3%), followed by 20 Ncm (23.1%), 30 Ncm (20.7%), 35 Ncm (18.7%), and 40 Ncm (23.3%). Importantly, all values for KSIII remained below 25%. Intra-group analysis revealed no statistically significant differences in torque loss rate among different tightening torques within the KSIII system (ANOVA, *p* > 0.05).

Independent *t*-test comparisons demonstrated that the KSIII system exhibited significantly lower torque loss rates than the TSIII system at every torque level tested (*p* < 0.001 for all torque values). On average, the torque loss rate of KSIII was approximately 2.5–3.0 times lower than that of TSIII, underscoring the superior mechanical stability of the KSIII abutment screw design under cyclic fatigue conditions.

## 4. Discussion

The present study evaluated the influence of tightening torque on screw stability in two implant systems. The principal finding was that the KSIII system consistently showed torque loss rates below 25%, regardless of torque level, whereas the TSIII system exhibited significantly higher loss rates, ranging from 36.5% to 51.8% depending on torque. This indicates that screw design exerts a profound effect on preload maintenance under cyclic loading.

### 4.1. Comparison with Literature

These results indicate that the KSIII abutment screw design allowed for more effective transfer of tightening torque into preload compared with the TSIII design, even before cyclic loading was applied. The higher torque loss in TSIII aligns with previous reports describing preload reduction due to frictional dissipation and settling [6,7,8,9,10]. Martin et al. [6] emphasized that friction accounts for most of the applied torque, limiting effective preload. Our results corroborate this, as TSIII demonstrated greater susceptibility to preload loss, particularly at lower torque levels. Conversely, the superior performance of KSIII parallels findings from studies on modified screw geometries and conical connections, which have been shown to improve load distribution and reduce loosening [14,15,16]. Hung et al. [15] reported that Morse taper systems achieved greater fatigue resistance than external hex connections, consistent with KSIII’s stable performance.

### 4.2. Tightening Torque and Retightening

Our data showed that for TSIII, torque loss was highest at 20 Ncm (51.8%) and lowest at 35 Ncm (36.5%). These findings align with recommendations that torque levels of 30–35 Ncm are generally optimal for abutment screw tightening [21]. Excessive torque (40 Ncm) did not further reduce loss and may risk microdamage, a concern previously highlighted by Bickford [26]. In contrast, KSIII demonstrated stable preload retention across all torque levels, suggesting less sensitivity to tightening variation. This finding implies that the KSIII system may provide a broader safety margin for clinical torque application. Retightening protocols remain clinically useful, particularly in systems prone to early preload loss [11,12]. However, the consistently low loss rates of KSIII suggest that its design inherently mitigates the need for frequent retightening.

### 4.3. Biomechanical and Clinical Implications

Preload stability is fundamental for resisting fatigue and ensuring long-term prosthesis function [27]. FEA studies have demonstrated that adequate preload reduces micromotion, distributes occlusal stresses more evenly, and prolongs component lifespan [28,29]. The smaller head diameter, narrower thread profile, and longer nominal length of KSIII screws likely contribute to improved stress distribution and reduced concentration at critical points, explaining the significantly lower torque loss observed. Clinically, this translates into fewer screw loosening events, reduced maintenance visits, and greater patient satisfaction. In addition to screw loosening, the risk of implant or abutment fracture should also be considered, as such complications can severely compromise long-term outcomes. Recent consensus guidelines emphasize preventive strategies, including appropriate abutment selection, torque control, and implant design, to minimize the risk of fracture and peri-implantitis [30,31]. Moreover, experimental studies have demonstrated that multi-directional cyclic loading produces higher fatigue stresses than uniaxial loading, thereby increasing the likelihood of implant fracture in clinically relevant conditions [22].

Screw loosening is particularly problematic in posterior single crowns and long-span fixed partial dentures, where non-axial forces and parafunction increase mechanical demand [2,4]. Previous clinical studies documented loosening as a common reason for prosthetic failure, with negative effects on peri-implant bone stability [3,5]. By maintaining preload more effectively, KSIII may mitigate such risks. Furthermore, contamination at the interface has been shown to alter removal torque values, with saliva and blood sometimes increasing and chlorhexidine reducing RTVs [20,21]. The robust stability of KSIII across torque levels suggests potential resilience even under variable intraoral conditions, though clinical validation is required.

### 4.4. Limitations and Future Directions

This study has limitations. The experimental setup simulated approximately 100,000 loading cycles, equivalent to only a short period of function. Longer-term studies are needed to replicate years of mastication. In addition, only two implant systems were examined, limiting generalizability. The in vitro environment cannot reproduce the complex oral milieu, including thermal cycling, humidity, and microbiological factors that may accelerate preload loss [23]. Furthermore, no microscopic surface characterization (e.g., SEM or TEM imaging) was performed after torque testing, which could have provided additional insight into wear patterns, deformation, or surface alterations of the abutment screws.

Future research should include extended cyclic loading with thermocycling and contamination protocols. Comparative studies of additional systems with varying materials, coatings, and geometries are needed. Advanced FEA modeling can provide further insight into stress distributions within screws of different designs [24,25]. Finally, prospective clinical trials are required to confirm that the biomechanical advantages demonstrated by KSIII translate into reduced screw loosening and improved implant survival in patients.

## 5. Conclusions

Within the limitations of this in vitro study, the following conclusions can be drawn:(1)The KSIII system consistently demonstrated lower torque loss rates (<25%) than the TSIII system (>35%), regardless of tightening torque, indicating superior mechanical stability of the implant–abutment complex.(2)For both systems, this torque level (35 Ncm) minimized preload loss while avoiding the increased loss observed at lower (20–25 Ncm) or excessively high (40 Ncm) values.(3)By maintaining preload more effectively, the KSIII system may reduce the risk of screw loosening, enhance prosthetic stability, and lower the frequency of maintenance visits in clinical settings.(4)Results are based on short-term fatigue testing and cannot fully replicate the oral environment. Long-term cyclic loading, thermomechanical aging, and clinical trials are required to confirm whether these biomechanical advantages translate into reduced complication rates in patients.

In summary, the KSIII implant system offers biomechanical and potential clinical advantages over the TSIII system, especially with a tightening torque of 35 Ncm. These findings provide evidence-based guidance for clinicians in optimizing screw tightening protocols and selecting implant systems to minimize screw loosening and enhance the long-term success of implant-supported restorations.

## Figures and Tables

**Figure 1 bioengineering-12-01131-f001:**
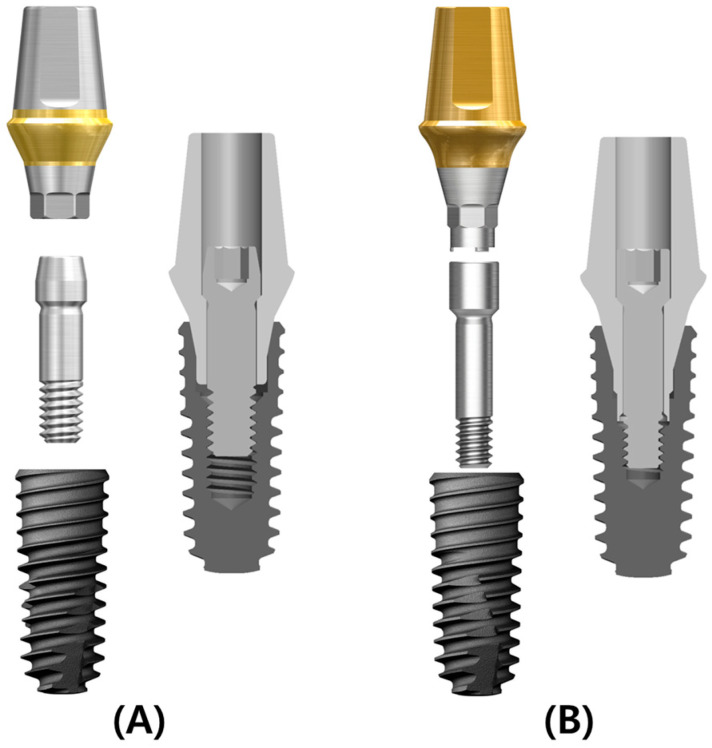
Components of the implant systems tested. (**A**) TSIII system with a conventional abutment screw (standard head and thread geometry); (**B**) KSIII system with a modified screw design (reduced head diameter, narrower thread profile, and increased nominal length).

**Figure 2 bioengineering-12-01131-f002:**
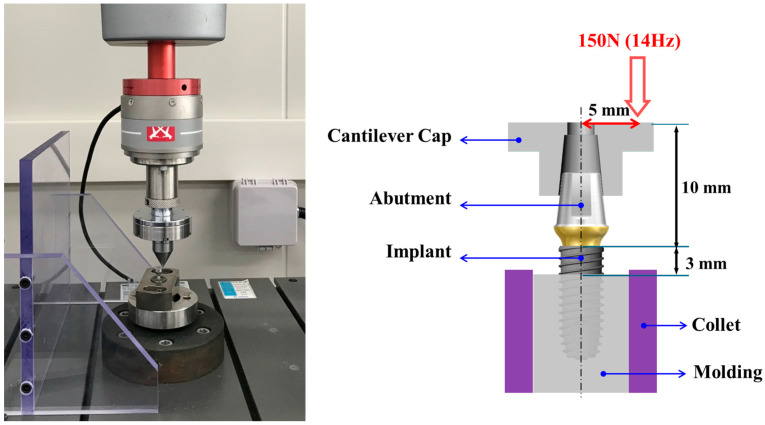
Universal testing machine setup for cyclic fatigue loading. Implant–abutment assemblies were mounted at a 30° angle relative to the implant axis and subjected to 150 N at 14 Hz for 100,000 cycles to simulate off-axis occlusal forces during mastication.

**Figure 3 bioengineering-12-01131-f003:**
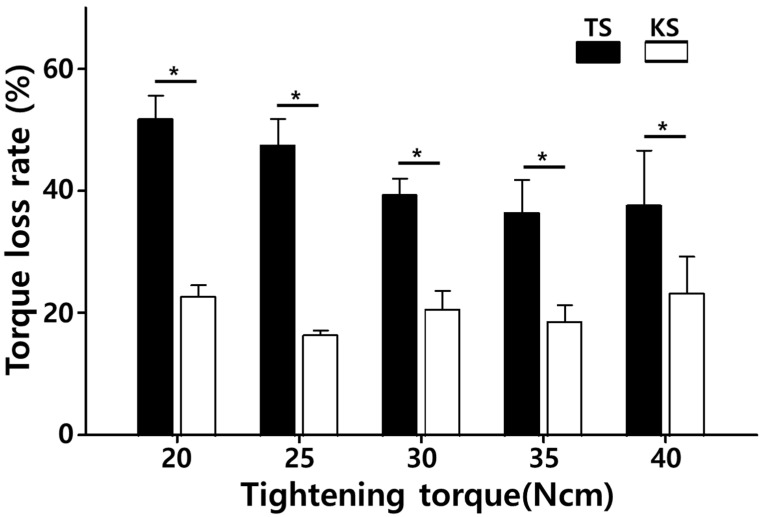
Mean torque loss rates (%) after cyclic loading (150 N, 14 Hz, 100,000 cycles) for TSIII and KSIII systems at different tightening torques (20–40 Ncm). Error bars indicate standard deviations. * indicates *p* < 0.05 compared with KSIII at the same torque level.

**Table 1 bioengineering-12-01131-t001:** Initial removal torque (T1, measured 5 min after tightening) for TSIII and KSIII systems at different tightening torque levels (20–40 Ncm). Values are presented as mean ± standard deviation (SD). An independent *t*-test showed that KSIII consistently exhibited significantly higher T1 values than TSIII across all torque levels (*p* < 0.05).

Tightening Torque (Ncm)	TSIII (Mean ± SD)	KSIII (Mean ± SD)	*p*-Value
20	16.4 ± 0.3	18.2 ± 0.3	0.002
25	19.3 ± 0.3	20.8 ± 0.3	0.004
30	23.1 ± 0.7	27.1 ± 0.2	<0.001
35	30.3 ± 0.3	33.1 ± 0.3	0.001
40	29.3 ± 1.1	32.2 ± 0.6	0.006

**Table 2 bioengineering-12-01131-t002:** Post-fatigue removal torque (T2, measured after cyclic loading of 150 N at 14 Hz for 100,000 cycles) for TSIII and KSIII systems at different tightening torque levels (20–40 Ncm). Values are presented as mean ± SD. KSIII maintained significantly higher T2 values compared with TSIII at all torque levels (*p* < 0.05, independent *t*-test).

Tightening Torque (Ncm)	TSIII (Mean ± SD)	KSIII (Mean ± SD)	*p*-Value
20	7.9 ± 0.2	14.0 ± 0.4	<0.001
25	10.1 ± 0.5	17.4 ± 0.2	<0.001
30	14.0 ± 0.6	21.5 ± 0.4	<0.001
35	19.2 ± 0.6	26.9 ± 0.5	0.002
40	18.1 ± 0.7	24.7 ± 0.8	0.003

## Data Availability

The datasets generated during and/or analyzed during the current study are available from the corresponding author on reasonable request.

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
