# Peer review of "Effect of Abutment Screw Design on Torque Loss Under Cyclic Fatigue Loading: A Comparison of TSIII and KSIII Implant Systems"

_bioengineering, 2025, doi:10.3390/bioengineering12101131_

Round 1

Reviewer 1 Report

Comments and Suggestions for Authors

The authors provided a clear and well-written manuscript of the effect of cyclic loading on screw losening of two implants systems. This information is important for the clinicians. Below are some suggestions for the authors to address before publication:

  • Line 100-101: The affirmation of "perform better" is too broad for this context. What would be considered better?
  • Line 115: A better description of the sample production is needed. Were the implants embedded in something? Was a holder holding the implants while performing the torque? Detailed information is needed.
  • Line 122: Did the same person perform the torque for all of the samples? If so, please add this information.
  • Line 134: Provide a reference for this sentence.

Author Response

Please find attached the file.

Reviewer 2 Report

Comments and Suggestions for Authors

Dear authors thanks for this research. I suggested some minor corrections.

Lines 100-101 must to be deleted. "However, based on screw design differences, it was anticipated that KSIII may perform better.". You can write. The null hypothesis has been tested again the hypothesis of differences, or something like this. You can not anticipate the results.

2.1. Implant Systems and Abutments. Please specify that implants have the same macro and micro-design (please confirm) and just different implant-abutment connection.

2.2. Sample Size and Grouping. How sample size has been calculated?

Fatigue Loading Protocol. Where the tests have been performed? Provate research center? University? Please specify the location.

Line 158 "As expected" can be deleted.

Lines 168-170 please move to the discussion section.  "These results indicate that the KSIII abutment screw design allowed for more effective transfer of tightening torque into preload compared with the TSIII design, even before cyclic loading was applied."

In the discuss, please report also risk of fracture including, if you want, this paper. Tallarico, M.; Lee, S.-y.; Cho, Y.-j.; Noh, K.-t.; Chikahiro, O.; Aguirre, F.; Uzgur, R.; Noè, G.; Cervino, G.; Cicciù, M. Prosthetic Guidelines to Prevent Implant Fracture and Peri-Implantitis: A Consensus Statement from the Osstem Implant Community. Prosthesis 2025, 7, 65. https://doi.org/10.3390/prosthesis7030065

Author Response

Please find attached the file.

Reviewer 3 Report

Comments and Suggestions for Authors

1.Please specify the material of each component shown in Figure 1, with particular     

   attention to the TSIII and KSIII components.  

2. Could you provide a reference for the torque testing method used? For example,

    is it based on ISO standards or other published literature?  

3. Could you explain the rationale for using a one-way ANOVA followed by a post-hoc 

    Tukey’s test to assess differences among torque levels? Are there any references

    supporting this approach?  

4. Was any analysis conducted to examine surface structural changes of the samples after

    the torque test, such as SEM or TEM imaging?

Comments on the Quality of English Language

Need a proofreading. 

Author Response

Please find attached the file.

Round 2

Reviewer 2 Report

Comments and Suggestions for Authors

Dear authors thanks to provide a revisited version of your research according to my suggestions.

Reviewer 3 Report

Comments and Suggestions for Authors

The author has revised the article in accordance with the reviewers' recommendations; therefore, I recommend that the article be accepted for publication.